# Validation of the Pathogenic Effect of *IGHMBP2* Gene Mutations Based on Yeast *S. cerevisiae* Model

**DOI:** 10.3390/ijms23179913

**Published:** 2022-08-31

**Authors:** Weronika Rzepnikowska, Joanna Kaminska, Andrzej Kochański

**Affiliations:** 1Neuromuscular Unit, Mossakowski Medical Research Institute, Polish Academy of Sciences, 02-106 Warsaw, Poland; 2Institute of Biochemistry and Biophysics Polish Academy of Sciences, 02-106 Warsaw, Poland

**Keywords:** SMARD1, IGHMBP2 gene, missense mutations, disease model, yeast *Saccharomyces cerevisiae*, CMT2S

## Abstract

Spinal muscular atrophy with respiratory distress type 1 (SMARD1) is a heritable neurodegenerative disease characterized by rapid respiratory failure within the first months of life and progressive muscle weakness and wasting. Although the causative gene, *IGHMBP2*, is well defined, information on *IGHMBP2* mutations is not always sufficient to diagnose particular patients, as the gene is highly polymorphic and the pathogenicity of many gene variants is unknown. In this study, we generated a simple yeast model to establish the significance of *IGHMBP2* variants for disease development, especially those that are missense mutations. We have shown that cDNA of the human gene encodes protein which is functional in yeast cells and different pathogenic mutations affect this functionality. Furthermore, there is a correlation between the phenotype estimated in in vitro studies and our results, indicating that our model may be used to quickly and simply distinguish between pathogenic and non-pathogenic mutations identified in *IGHMBP2* in patients.

## 1. Introduction

Spinal muscular atrophy with respiratory distress type 1 (SMARD1) represents one of the most severe neonatal conditions diagnosed in newborns hospitalized in neonatal intensive care units. Typically, the affected children manifest with acute respiratory insufficiency, which is accompanied by diaphragm paralysis, muscle weakness, heart arrhythmia, and other anatomical disturbances [1,2,3]. In general, the patients die before their second year of life; however, some late-onset SMARD1 cases have been reported in the literature [4,5,6,7].

SMARD1 results from mutations in *IGHMBP2* (Immunoglobulin Mu-Binding Protein 2) gene [8] and is inherited in an autosomal recessive trait. Thus, accurate diagnosis of SMARD1 requires the detection of two pathogenic mutations occurring in a trans configuration within the causative gene. In patients with severe respiratory distress harboring two identical (homozygous) or two different (compound heterozygosity) nonsense *IGHMBP2* mutations diagnosis of SMARD1 is not questionable. However, the process of diagnosis is complicated when a single *IGHMBP2* mutation without a documented pathogenic status is detected, especially in newborns manifesting an atypical course of SMARD1. Theoretically these children may be considered as carriers of a single *IGHMBP2* mutation (the second variant being potentially harmless polymorphism) manifesting with a separate clinical entity, other than SMARD1 but with a similar clinical manifestation. In those patients the coexistence of *IGHMBP2* polymorphism or a variant of unknown pathogenic significance with a *IGHMBP2* mutation may result in a false clinical diagnosis. In fact, in decisions concerning the implementation of intensive therapy in newborns the results of *IGHMBP2* gene testing may be essential. In case of rapidly worsening neonatal disease in particular, the time for an assessment of the pathogenic effect of *IGHMBP2* gene variants is very limited. Moreover, after the death of a SMARD1-affected newborn, the *IGHMBP2* variants detected in the child will be further used in genetic counselling for parents and further generations. Thus, the classification of an *IGHMBP2* allele has important clinical relevance. It should be noted that mutations in the *IGHMBP2* gene do not always result in SMARD1 but may also lead to a milder form of neurodegeneration, namely Charcot-Marie-Tooth type 2S (CMT2S) disease [9]. In CMT2S, typically, the first symptoms are observed in childhood and usually include the progressive weakness and wasting of the distal muscle [9,10,11]. Although the disease is not usually life-threating, it may lead to the severe disability [9].

Traditional methods of evaluation of the pathogenic status of *IGHMBP2* sequence variants may be used in a limited range [12,13]. SMARD1 pedigrees are usually too small to be serviceable for an analysis of mutation segregation. Additionally, the patients predominantly do not reach 5 years of age; therefore, the medical documentation from them is rather scarce. Moreover, the *IGHMBP2* gene is highly polymorphic and various bioinformatics predictions often provide conflicting results. The evaluation of pathogenic status of the *IGHMBP2* sequence variants is also hindered by poor knowledge about the functioning of IGHMBP2. It is well established that this protein is a 993 amino acid (aas) ATP-dependent RNA/DNA helicase [14,15]. It consist of a helicase domain, a single-stranded (ss) RNA and DNA binding R3H domain, and an AN1-type zinc finger motif (zf-AN1) [16,17]. It may participate in transcription [18,19,20], mRNA splicing [14], and rRNA and tRNA metabolism and translation [15,21,22], but the exact requirement of IGHMBP2 in particular processes is still unexplained.

IGHMBP2 is well-conserved, and its homologs are found in many model organisms, from *Saccharomyces cerevisiae* to mice [17]. The *S. cerevisiae* homolog of IGHMBP2 is helicase Hcs1 (DNA HeliCaSe). Although, it is shorter than IGHMBP2 (possessing 683 aas) and lacks the C-terminal domains (R3H and zf-AN1), it is well conserved: the full-length proteins share approximately 23% identity and 38% similarity, but the helicase domains are 35% identical and 57% similar [17].

Taking into account the overall complications in the classification of *IGHMBP2* alleles, good, accessible, and reproducible systems for testing newly identified sequence variants are very valuable. One of the simplest and most willingly used models is the yeast *Saccharomyces cerevisiae*. This unicellular organism is broadly used in researching the wide spectrum of human neurodegenerative disorders including hereditary spastic paraplegia [23], Alzheimer’s, and Parkinson’s diseases [24,25,26], Huntington’s disease [27], chorea-acanthocytosis [28], and neuropathies [29] as well as in other disorders, such as mitochondrial diseases [30] and copper metabolism disorders [31,32]. The potential of yeast for use in the validation of the pathogenicity of rare disease variants [32,33] in the assessment of the mechanisms of allele dysfunction [34,35] and in basic science [36] has been proved. It also offers a cheap, easy, and rapid platform for the high-throughput screening of genetic and chemical suppressors that could give rise to potential therapeutics [37,38]. Thus, yeast is a useful, practical, and versatile tool for use in the study of human diseases and mutations associated with inherited disorders.

In this study, we asked whether the pathogenic effect of *IGHMBP2* mutations may be evaluated using a functional approach. We generated the *S. cerevisiae* platform to study the functionality of *IGHMBP2* mutants in vivo. We identified a previously undescribed growth phenotype of the *hcs1*Δ mutant yeast strain that lacks the homolog of IGHMBP2, the Hcs1 protein. We also showed that a full-length but not truncated version of IGHMBP2 suppresses this unfavorable phenotype. In contrast to benign mutations, most pathogenic mutations impair this suppression. These results demonstrate the accuracy of our model and its great potential for use in studying the rare variants of *IGHMBP2* found in patients and in the diseases caused by *IGHMBP2* mutations.

## 2. Results

### 2.1. The HCS1 Deficient Strain Is Not Hypersensitive to 4-Chlorophenol on Solid YPD Medium

SMARD1 disease is very rare, and additionally, the associated *IGHMBP2* gene is highly polymorphic, thus the interpretation of the results of gene sequencing obtained from patients may be challenging. To facilitate diagnosis, we generated a simple and reproducible model that allowed us to study in vivo the effect of the missense mutations found in the *IGHMBP2* gene. We chose yeast *Saccharomyces cerevisiae* as a convenient platform due to its low-cost culturing, short generation time, and high genetic tractability. As yeast possesses the Hcs1 protein, the homolog of human IGHMBP2, we constructed our model using *hcs1*Δ strains lacking the *HCS1* gene. We hypothesized that this deletion mutation may mimic the loss-of-function mutations of *IGHMBP2* usually observed in SMARD1 patients and thus is most suitable for further research. To generate the practical model, we needed to find an easy to analyze and repeat phenotypes. The most rapid and useful one is the defective growth of yeast cells on solid plates under specific conditions. In the Saccharomyces Genome Database (SGD; https://www.yeastgenome.org/; accessed in 6 July 2020) we found that the *hcs1*Δ strain is hypersensitive to 4-chlorophenol (4-CP) [39]. To test usefulness of this phenotype we introduced the *hcs1*Δ mutation into BY4741 and BY4742 backgrounds. Several independent clones bearing a desired deletion in each background were plated on solid YPD medium containing the 4-CP. We were able to observe only a slightly slower growth of 3, 4, 5, and 7, but not the 1 and 8 mutant clones compare to wild-type (WT) in the BY4741 background. None of the mutant clones in the BY4742 background were more sensitive to 4-CP than the wild-type strain, even in a very high concentration (2 mM) of this chemical (Figure 1a). Most likely, therefore, the *hcs1*Δ mutant is not sensitive to 4-CP, at least on solid medium. To test if the observed differences were really caused by the lack of *HCS1* gene in the BY4741 background, we constructed a plasmid containing the *HCS1* gene under native promoter and terminator, and tested if it was able to complement the growth defect of the *hcs1*Δ mutant (Figure 1b). Surprisingly, the strains expressing the *HCS1* gene from plasmid were slightly more sensitive to 4-CP than the strains with an empty vector (Figure 1b). Thus, the observed 4-CP hypersensitivity of the *hcs1*Δ mutants was not *HCS1*-dependent, however, an additional copy of *HCS1* confers sensitivity to 4-CP.

### 2.2. The Deletion of HCS1 Leads to Hypersensitivity to Cycloheximide

As the *hcs1*Δ strain did not appear to have simple growth phenotypes, we decided to construct double mutants, without the *HCS1* gene and the other gene, to find a combination that results in a stronger phenotype in comparison with each single mutant. Based on the genetic and physical interactions data available in SGD, we selected several candidate genes to delete in the *hcs1*Δ strain, namely *CCR4, DOA1* and *MAF1*. The *CCR4* gene encodes Ccr4, a component of the CCR4-NOT transcriptional complex [40] that physically interacts with Hcs1, while the other interactors are genetic (SGD accessed in July 2020). *DOA1* encodes the Doa1 protein involved in ubiquitin-mediated protein degradation [41], while *MAF1* is a negative regulator of RNA polymerase III [42], which is responsible for tRNA genes transcription. Additionally, based on the Pfam database (http://pfam.xfam.org/; accesses on 8 July 2020), we selected yeast proteins that have the R3H domain. In human IGHMBP2, this domain enhances both RNA binding and the ATPase activity of the IGHMBP2 helicase domain [16], but it is absent in yeast Hcs1. Thus, we hypothesized that other proteins possessing the R3H domain may interact with Hcs1 to facilitate its function in cells. Therefore, the *FAP1*, *RBS1*, and *SQS1* genes were selected to study the effect of their deletion in a strain already bearing the *HCS1* gene deletion. These genes encode, respectively, the following proteins: Fap1, which confers rapamycin resistance [43]; Rbs1 which is involved in assembly of the RNA polymerase III complex [44] and Sqs1, which stimulates ATPase and the helicase activities of Prp43, the RNA helicase [45]. After obtaining the double mutants, we tested their phenotypes on different solid media, such as the YPEG medium with a non-fermentable carbon-source, synthetic complete (SC) and synthetic minimal (SD) medium, and on YPD medium under various conditions, such as different temperatures (37 °C and 13 °C), and in the presence of various ions (Ca^2+^, Cu^2+^, Mg^2+^, Na^+^ and Zn^2+^) and chemicals (benomyl, caffeine, dimethyl sulfoxide (DMSO), 1,4-Dithiothreitol (DTT), ethanol, boric acid, hydroxyurea (HU), myriocin, sodium dodecyl sulfate (SDS) and cycloheximide (CHX)). The testing conditions were selected based on the phenotypes described in the SGD for the chosen single mutants. The obtained results are presented in Appendix A. We did not observe a significant impairment of the growth of the double mutants compared to single ones, but we did observe that even the single *hcs1*Δ mutant presented a clear growth defect on the medium containing CHX, especially in the BY4741 background (Figure 2 and Appendix A). We also confirmed that the observed growth defect depends on the absence of *HCS1,* as the *HCS1* gene provided on the plasmid was able to complement the observed phenotype (Figure 2). The sensitivity of the *hcs1*Δ strain to CHX suggests that Hcs1 is probably involved in translation similarly to IGHMBP2.

### 2.3. Full-Length Human IGHMBP2 Suppresses the Hypersensitivity of hcs1Δ Mutant to Cycloheximide

The helicase domains of Hcs1 and IGHMBP2 proteins are well conserved (35% identity and 57% similarity [17]; thus, we speculated whether human protein can, at least partially, fulfill the function of Hcs1. We constructed several plasmids to answer this question. First, we used single and multicopy vectors with various promoters giving different levels of expression of the selected genes (from the weakest to the strongest: *CYC1*, *ADH1*, *TDH3*). The yeast protein is shorter than the human one and it has no C-terminal domains (R3N and ZF-AN2; Figure 3a), so we constructed not only plasmids containing the full-length cDNA of *IGHMBP2* but also a shortened version encoding only the helicase domain (aas 1-652). We used a phenotype of CHX hypersensitivity to test the functionality of human IGHMBP2 in yeast cells. Full-length *IGHMBP2* expressed from multicopy plasmid with a very strong promoter partially complemented the lack of *HCS1* in the yeast cells. This effect was not observed for the truncated version (Figure 3b), even though the level of shortened protein was much higher compared with full-length IGHMBP2 (Figure 3c). This confirms previous reports that C-terminal domains are important for IGHMBP2 functioning [46] and that the nonsense mutations that result in the truncation of IGHMBP2 lead to the disease phenotype [9,47].

### 2.4. Selection of Missense Mutations in IGHMBP2 for Validation of Our Yeast System

Our results showed that wild-type *IGHMBP2* encodes a functional protein in yeast cells, indicating that the effect of mutations found in patients on the functionality of the IGHMBP2 protein can be directly studied in the human gene. We focused on missense mutations. Most of the pathogenic substitutions found in SMARD1-affected patients are located in the helicase domain and usually lead to a reduction in the helicase and/or ATPase activity of the IGHMBP2 protein tested in vitro [15,48]. Thus, we focused on changes in the helicase domain. For further analysis, we selected 11 *IGHMBP2* variants. An allele containing the nonsense mutation c.163C > T p.Gln55* was used as a control, as it leads to the production of residual protein, which could not be functional. Two other mutations, namely c.223G > A p.Ala75Thr and c.823A > G p.Ile275Val were described as a benign in the ClinVar database (https://www.ncbi.nlm.nih.gov/clinvar/ (accessed on 19 May 2021). c.151C > G p.Gln51Glu was reported with conflicting interpretations of pathogenicity as benign, likely benign and uncertain significance in the ClinVar database, thus we included it to our study as a testing object. Four other mutations have been reported in the literature as being associated with SMARD1, namely c.595G > A p. Ala199Pro [1], c. 767C > G p.Ala256Gly [49], c.1082T > C p.Leu361Pro [5,47,50,51,52,53], and c.1794C > A p.Asn598Lys [1]. However, c.595G > A p. Ala199Pro, c. 767C > G p.Ala256Gly and c.1794C > A p.Asn598Lys are indicated as the variants of an uncertain significance (VUS) in the ClinVar database. We also included a mutation, so far linked only to CMT2S, c.734A > G p.Asn245Ser [9] but reported as a variant with uncertain significance in the ClinVar database. Additionally, it was reported in heterozygote, while the second allele was not identified [9], thus it was thought that testing its pathogenicity would be important. Additionally, we chose mutations that, depending on second allele, may result SMARD1 or CMT2S phenotypes, namely c.1478C > T p.Thr493Ile [5,53,54,55,56] and c.1738G > A p.Val580Ile [8,9,48,57] (Figure 3a; see also Table 1). Three of the above selected mutations (c.1082T > C p.Leu361Pro, c.1478C > T p.Thr493Ile and c.1738G > A p.Val580Ile) had been well characterized in in vitro studies [15,48], thus allowing us to compare the results obtained in different experimental systems.

### 2.5. Pathogenic Mutations in IGHMBP2 Gene but Not in HCS1 Results in a Loss-of-Function in Yeast Model

To confirm the accuracy of our model, we introduced the previously selected mutations (Figure 3a) into the *IGHMBP2* gene and tested whether this results in the loss of function of IGHMBP2, manifested as an inability to restore the growth of yeast *hcs1*Δ cells on a CHX-containing medium. As expected, the alleles of *IGHMBP2,* being the benign mutations c.223G > A p.Ala75Thr and c.823A > G p.Ile275Val, behaved as wild-type allele. Similarly, c.151C > G p.Gln51Glu did not impaired the functioning of the encoded IGHMBP2 protein in the yeast cells. In contrast, the nonsense mutation resulting in Gln55* completely diminished the ability of *IGHMBP2* to improve the growth of *hcs1*Δ mutant yeast cells on the CHX-containing medium. Six of the seven other studied pathogenic mutations resulted in the impairment of IGHMBP2 protein functionality but with varying degrees of intensity. Three substitutions Ala199Pro, Asn245Ser and Asn598Lys, similarly to Gln55*, abolished the ability of IGHMBP2 to reduce the hypersensitivity of the *hcs1*Δ mutant to CHX. The allele encoding IGHMBP2-Val580Ile exhibited a weak ability to suppress yeast mutant growth defects in a lower (0.4 µg mL^−1^) CHX concentration, but this effect was not observed in a higher (0.6 µg mL^−1^) CHX concentration. The alleles *IGHMBP2-Ala256Gly* and *-Leu361Pro* improved the *hcs1*Δ growth in a lower concentration of CHX quite well, but they were inactive in a higher CHX concentration (Figure 3d). Only the *IGHMBP2-Thr493Ile* allele suppressed the growth defect similarly to wild-type *IGHMBP2* (Figure 3d); however, a previous report indicates that this mutant exhibits neither a reduction in ATPase activity nor helicase activity, only a lower RNA/DNA binding capacity was observed [15]. The suppression efficiency was not dependent on the protein level as the amount of the mutant proteins in all studied cases was similar (Figure 3e). In addition, we chose four amino acids residues conserved between human IGHMBP2 and, yeast Hcs1 proteins, which are substituted in patients (Ala199Pro, Thr493Ile, Val580Ile and Asn598Lys; in yeast, respectively: Ala215Pro, Thr525Ile, Val616Ile and Asn634Lys; Figure 3a and Appendix A) and studied if their changes impair Hcs1 protein function. Only the replacement of Asn598 to Lys (Asn634Lys in yeast) led to the impairment of Hcs1 functionality in cells (Figure 3f), indicating that this residue is important for the function of both (human and yeast) homologous proteins.

## 3. Discussion

Next-generation sequencing (NGS) is an excellent tool for use in the diagnosis of patients suffering from genetic disorders. However, massive genome sequencing reveals many rare gene variants of unknown clinical relevance. A determination of the disease-causative mutations is not always easy due to the broad spectrum of clinical symptoms, various ages of disease onset, a lack of data regarding the natural course of the disease in some cases and, in particular, the lack of easy and standardized functional tests of altered gene products. Missense mutations, leading to the substitution of amino acids resides in proteins, are especially hard to classify. The problem is particularly disturbing with genes that are highly polymorphic. According to the ClinVar database (accessed on 3 August 2022), the vast majority of *IGHMBP2* mutations are VUS (561 cases). Likely benign and benign variants (412) are overwhelming the smallest category of pathogenic (88) and likely pathogenic (49) sequence variants. The most challenging molecular diagnostics category of variants, i.e., VUS, represent potentially pathogenic or benign gene alterations. Through in vivo studies of IGHMBP2 protein functionality, the VUS category for *IGHMBP2* may be reduced by the shift to pathogenic or benign variants. Determining the distinction between mutations from polymorphisms and those from benign changes is very challenging and, if unsuccessful, may result in false positive or negative diagnoses. Thus, gene variant classification is fundamental for genetic testing. This clearly shows that simple and rapid methods for testing the functional defects caused by individual gene mutations are highly needed. The mutations within the *IGHMBP2* gene usually occur in the compound heterozygous state. Thus, in patients, there is no possibility to analyze separately the impact of a single *IGHMBP2* mutation. The assay, presented by us, provides a unique opportunity to analyze the effect of single *IGHMBP2* mutations instead of the combined impact of two different mutations occurring in heterozygotes.

SMARD1 represents a challenging clinical situation in terms of molecular diagnostics. In fact, due to the limited prognostic value of the clinical symptoms observed in some newborns, the results of *IGHMBP2* gene testing may determine the clinical management. Due to the rapid course of the disease, there is no time for clinical observation, as there is with other chronic, slowly progressing heritable disorders. The results of *IGHMBP2* analysis are crucial not only for genetic counseling but also, above all, for the appropriate intensive or palliative neonatal treatment. Therefore, we investigated the potential of a yeast-based model in functional studies of IGHMBP2 variants. In this study, we generated a platform for easily assaying the mutations found in the *IGHMBP2* gene. Using this, we tested 11 *IGHMBP2* variants and obtained good agreement with other data available in the literature and in databases (Table 1). The mutations, referred to as benign in the ClinVar database, resulting in the Ala75Thr and Ile275Val substitutions did not, in fact, have any impact on yeast growth, as it did for the wild-type *IGHMBP2*. Similarly, Gln51Glu, reported as benign, likely benign or of uncertain significance, did not influence the functioning of IGHMBP2 in yeast cells, suggesting that these changes are not pathogenic. The nonsense mutation resulting in the Gln55* caused a complete loss-of-function and totally abolished the ability to improve the growth of *hcs1*Δ cells on the CHX-containing medium. The variants p.Ala199Pro, p.Asn245Ser, and p.Asn598Lys caused the same effect. However, the alleles encoding p.Ala199Pro and p.Asn598Lys, reported as VUS were found in compound heterozygote patients with severe SMARD1 [1]. Individuals with p.Ala199Pro/p.Ser539_Tyr541del and one with p.Arg147*/p.Asn598Lys presented intrauterine growth retardation, a weak cry and feeding problems after birth, respiratory insufficiency at the age of 3 months and 8 weeks, and sudden death during the night at the age of 19 months and 15 months, respectively [1]. The second patients with p.Arg147*/p.Asn598Lys (the younger sister of the previously described individual) developed acute respiratory distress at the age of 2 months and died at the age of 40 months [1]. Clinical manifestations support our data that Ala199Pro and Asn598Lys substitutions lead to a strong impairment of IGHMBP2 functioning.

Although in our system p.Asp245Ser resulted in a strong phenotype, clinically, it was found that in CMT2S patients it manifested with rather mildly expressed clinical symptoms; at the age of 5 years, gait difficulty was observed, and at the age of 41, the patients were able to walk with ankle-foot orthosis [9]. However, in this case, the second allele of *IGHMBP2* was not identified [9]. The coexistence of the different variants seems to be important for the clinical manifestation of individual mutations. For example, *IGHMBP2-Val580Ile* in a homozygote results in a severe SMARD1 phenotype [8,48], while it was found in a heterozygote in CMT2S patient [9]. The dysfunction of the protein functioning with Val580 substitution was confirmed by an in vitro study, where no ATPase or helicase activity was observed [48]. The replacement of Val580 residue by isoleucine with a longer side chain likely destabilizes the helicase domain [9]. Furthermore, in our test, this variant was severely impaired. Thus, the presence of second sequence variant may modify the general clinical picture of the disease, even if one mutations is thought to be very strong. The *IGHMBP2-Asp245Ser* variant may lead to severe dysfunction of the IGHMBP2 protein, especially as it is predicted to cause protein instability, leading to a loss-of-functional protein [9], but in combination with other, more functional variants the final clinical phenotype may be relatively mild.

In our model, the visible defects in the functioning of the Ala256Gly and Leu361Pro IGHMBP2 variants are present only in higher CHX concentrations, suggesting that these variants, at least partially, maintain their function. This result is concurrent with the data obtained from clinical examination. The phenotype of individuals expressing *IGHMBP2-Leu361Pro* is strongly dependent on the second allele, and a broad spectrum of SMARD1, from severe to a milder form, is observed [5,47,50,51,52,53] (Table 1). Data concerning the effect of *IGHMBP2-Ala256Gly* allele are scanter [49,58], but it seems that, similar to the *IGHMBP2-Leu361Pro* allele, the functionality of the second allele determines the clinical picture: the second weaker mutation may result in a milder phenotype, while the stronger may lead to severe disease.

The expression of *IGHMBP2-Thr493Ile* suppresses the growth defect of *hcs1*∆ similar to a wild-type *IGHMBP2*. This is in line with an in vitro study that indicated that the p.Thr493Ile mutant protein exhibits neither a reduction in ATPase activity nor helicase activity, and only a decrease in RNA/DNA binding capacity was observed [15]. Structural analysis reveals that Thr493 residue is located away from the nucleotide-binding site and from any helicase motifs. Moreover, the side chain of Thr493 does not make significant contact with neighboring residues; thus, it is not known how the Thr493Ile substitution influences the conformation and functionality of IGHMBP2 [16]. It is proposed that Thr493Ile substitution may lead to a reduction in the protein level. Even in heterozygous carriers of the *IGHMBP2-Thr493Ile*, the level of IGHMB [5,16] decreased. In addition, the p.Thr493Ile variant of the IGHMBP2 protein had a tendency to form high-molecular-weight aggregates and to spontaneously degrade [5]. Although, in our system, we did not observe a reduced level of p.Thr493Ile mutant, it does not ruled out that scenario. The level of IGHMBP2 protein seems to be an important factor in the overall clinical manifestation of the disease. Some of the data even suggest that the severity of symptoms is correlated with the IGHMBP2 protein level [9]. However, this data must be considered with caution, as an intra-familiar variability of phenotypes was described [10,56], suggesting the presence of other disease modifiers. Importantly, it seems that for any pathological symptoms to be observed, the amount of IGHMBP2 protein needs to be below a certain threshold. In some healthy heterozygous carriers, a reduction in the IGHMBP2 protein level was also observed [5]. Overall, this may explain the visible variability of the phenotypes associated with Thr493Ile substitution (see Table 1). If the given mutation does not severely impair the functionality of the IGHMBP2 protein itself but reduces its level, the symptoms strongly depend on the strength of the second allele. For example, a heterozygous *IGHMBP2-Thr493Ile/IGHMBP2-Leu361Pro* combination (where the *IGHMBP2-Leu361Pro* variant is at least partially functional in our test) results in a milder phenotype than when the *IGHMBP2-Thr493Ile* allele is combined with variants with nonsense *IGHMBP2* mutations (causing p.Arg788* or p.Cys496*) [5,54].

Importantly, the model presented allowed us to test the functionality of the mutant IGHMBP2 protein in vivo in a cellular environment. To date, the effect of selected pathogenic mutations on IGHMBP2 functioning has been studied only in vitro [15,48] or based on structural study [9,16]. Naturally, proteins do not function as a single molecule, but, together with their partners, their activity may be limited to specific locations and situations. In vitro studies are usually focused on individual aspects of protein behavior and do not test the broad spectrum of factors that determinate protein performance. Of course the cellular environment of yeast is not the same as in mammals, however it still offers a more natural milieu for protein study.

Our model, as all tools used for the classification of individual variants of IGHMBP2, also possesses some limitations, as illustrated well by the IGHMBP2-Thr493Ile example. Specifically, we cannot predict the clinical severity and course of the disease based only on the characteristics of a particular allele. From all tested mutations, only *IGHMBP2-Val580Ile* was described in a homozygote and all other mutations were present in compound heterozygotes (Table 1). Hence, the exact contribution of each variant to the phenotype cannot be clearly defined. Furthermore, as mentioned above, the intra-familiar variability of phenotypes suggests the presence of factors that may modify the clinical course of the disease. The mouse genetic factors on chromosome 13 influenced the onset and progression of the disease in a SMARD1 mouse model described in [21,59,60]. Note that similarly protective allele also in humans cannot be excluded, and this additionally complicates the overall picture of the disease. Taken altogether, our model seems to be serviceable, especially for loss-of-function mutations within the *IGHMBP2* gene. The mutations acting in other modes may remain undetectable in our assay, or may even generate false-negative results. In these terms, an approach integrating all of the data (clinical, genealogical, bioinformatics, and functional) should be taken into consideration when the pathogenic effect of *IGHMBP2* gene variants is studied. However, particularly for non-recurrent *IGHMBP2* sequence variants lacking detailed phenotype characteristics, the yeast-based model proposed by us may play a leading role in the validation of the pathogenic effect. To conclude, our study provides clear evidence that, at least in case of ambiguous data concerning the pathogenic effect of *IGHMBP2* variants, functional studies based on yeast-model proposed by us may be serviceable in the clinical management of SMARD1-affected patients.

It should be noted that our platform not only allows for the testing of missense mutations but also represents a broad spectrum of utility. The *hcs1*Δ growth phenotype characterized by us could be used in rapid, high-throughput screening for chemical or genetic suppressors, which may result in new therapeutic possibilities, as it seems that the *hcs1*Δ strain may mimic the loss-of-function mutations in *IGHMBP2* found in patients. Moreover, it may be used for basic science research, especially that focusing on finding the molecular basis of pathogenicity and identifying the cellular pathways that require a functional IGHMBP2 protein. Human *IGHMBP2* could partially suppress the unfavorable phenotype of mutant *hcs1*Δ cells, which suggests that at least part of the functions and molecular partners are conserved between the yeast and human proteins. It also indicates that IGHMBP2 is involved in fundamental processes conserved from, at least, yeast to humans. We do not understand the molecular mechanisms underlaying CHX sensitivity phenotype of *hcs1*Δ mutant used in our model, as little is known about the function of both IGHMBP2 and Hcs1 proteins. Nevertheless, the construction of our model and the results provided by it show that it may be serviceable in the future to assess the pathogenic effect of *IGHMBP2* gene sequence variants. However, our study encompasses only a small representation of *IGHMBP2* gene variants. To be sure that the assay proposed by us can be applied to all *IGHMBP2* sequence variants, more SNPs need to be tested.

Altogether, our results demonstrate the utility of our system and show its promising potential for directly studying human *IGHMBP2*, its mutations, and its encoding protein in yeast cells.

## 4. Materials and Methods

### 4.1. Strains, Media, and Growth Conditions

*Escherichia coli* strains XL1-Blue were used for plasmid propagation. The yeast *Saccharomyces cerevisiae* strains used in this study are listed in Table 2.

Gene disruptions were performed by the transformation of yeast cells with a PCR product containing the *NatMX4* or *KanMX6* selection cassette, flanked by 50 bp of homology to the 5′ and 3′ regions of the locus of interest. The disruption of selected genes were confirmed by PCR using primers flanking the regions of interest. Yeast were grown at 28 °C in YPD medium (1% yeast extract, 2% peptone, 2% glucose), YEPG medium (1% yeast extract, 2% peptone, 2% ethanol, 3% glycerol), minimal synthetic medium (SD) (0.67% yeast nitrogen base with ammonium sulfate without amino acids, 2% glucose with desired supplements (uracil, amino acids)), or in complete synthetic medium (SC) (0.67% yeast nitrogen base with ammonium sulfate without amino acids, 2% glucose with complete supplement mixture (CSM)) either solid or liquid. For growth tests, cells were cultured overnight in liquid YPD, SC-leu, or SC-ura medium and were resuspended to OD600 ≈ 1 in the appropriate media. A total of 10 × serial dilutions were prepared. Three (YPD medium) or five (SC, SD or YEPG media) microliters of each dilution was spotted on appropriate plates (as indicated) and supplemented as indicated. Plates were incubated at 28 °C for the indicated number of days.

### 4.2. Plasmids

The plasmids used in this study are listed in Table 3.

The plasmids containing *HCS1* open reading frame (ORF) with native promoter and terminator sequences (chromosome XI coordinates: 405091-407465) were generated by amplification of the indicated fragment using genomic DNA from BY4741 as a template and primers providing restriction sites (EagI at 5′-end and HindIII 3′-end) and ligation of the obtained sequence with pRS315 or pRS416 vectors. Plasmids p415-P*_CYC1_*-*IGHMBP2*, p415-P*_ADH1_*-*IGHMBP2*, p415-P*_TDH3_*-*IGHMBP2*, p425-P*_TDH3_*-*IGHMBP2*, p415-P*_CYC1_*-*IGHMBP2*_652, p415-P_ADH1_-*IGHMBP2*_652, p415-P*_TDH3_*-*IGHMBP2*_652, and p425-P*_TDH3_*-*IGHMBP2*_652 were obtained by amplification of the full-length or truncated (encoding aas 1-652 of *IGHMBP2*) versions of *IGHMBP2* ORF using pCMV3-IGHMBP2 (HG 18840-UT; Sino Biological, Beijing, China) as a template and primers providing restriction sites (HindIII 5′-end and SalI 3′-end) and additional codon STOP (UAA) for *IGHMBP2*_652 and ligation of the obtained sequences into p415-P*_CYC1_*-, p415-P*_ADH1_*, p415-P*_TDH3_*, and p425-P*_TDH3_* vectors. The c.643G > C p.Ala215Pro, c.1574CG > TC p.Thr525Ile, c.1846G > A p.Val616Ile and c.1902C > A p.Asn634Lys mutations in *HCS1,* and the c.151C > G p.Gln51Glu, c.163C > T p.Gln55*, c.223G > A p.Ala75Thr, c.595G > A p.Ala199Pro, c.734A > G p.Asn245Ser, c. 767C > G p.Al256Gly, c.823A > G p.Ile275Val, c.1082T > C p.Leu361Pro, c.1478C > T p.Thr493Ile, c.1738G > A p.Val580Ile and c.1794C > A p.Asn598Lys mutations in the *IGHMBP2* gene were introduced by site-directed mutagenesis.

### 4.3. Site-Directed Mutagenesis

Site-directed mutagenesis was performed on the pRS315-*HCS1* and p425-P*_TDH3_*-*IGHMBP2* plasmids, using the Mut Express II Fast Mutagenesis Kit V2 (Vazyme Biotech Co., Ltd., Nanjing, Jiangsu, China) in accordance with the manufacturer’s protocol. Primers for mutagenesis were designed using PrimerX on-line software (https://www.bioinformatics.org/primerx/ (accessed on 20 May 2021). The primer sequences are listed in Appendix A. The presence of mutations within constructed plasmids was verified using the Sanger sequencing method.

### 4.4. Western Blot Analysis

Yeast cells were grown overnight at 28 °C in SC-leu medium. The following day, the cultures were refreshed and incubated for 4 h at 28 °C to exponential growth-phase. Then, they were placed on ice and PMSF, aprotynin, sodium aside, and sodium fluoride were added to the final concentration of 1 mM, 2 µg mL^−1^, 10 mM, and 10 mM, respectively. Protein extracts were prepared by disrupting cells with an acid-washed glass bead in CelLytic™ Y Cell Lysis Reagent (Sigma-Aldrich, Saint Louis, MO, USA) supplemented with 1,4-Dithiothreitol (DTT) and Protease Inhibitor Cocktail (Sigma-Aldrich). After centrifugation, supernatant was collected and 2× sample buffer (62.5 mM Tris-HCl pH 6.8; 2% SDS; 25% glycerol; 0.01% bromophenol blue; 5% 2-mercaptoethanol) was added. The samples were analyzed by standard western blotting using rabbit polyclonal anti-IGHMBP2 (Novus Biologicals, Littleton, CO, USA) or mouse monoclonal anti-Pgk1 (Abcam, Cambridge, MA, USA) antibodies and secondary anti-rabbit IgG or anti-mouse IgG horseradish peroxidase (HRP)-conjugated antibodies (Sigma-Aldrich), followed by enhanced chemiluminescence (Advansta, San Jose, CA, USA).

## Figures and Tables

**Figure 1 ijms-23-09913-f001:**
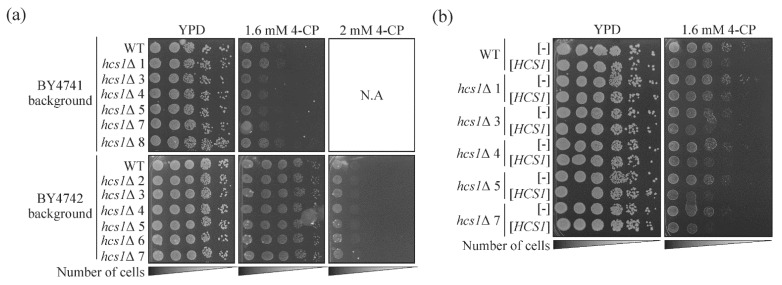
The influence of *HCS1* gene deletion and its ectopic expression on sensitivity to 4-chlorophenol (4-CP). (**a**) The strain *hcs1*Δ is not hypersensitive to 4-CP on a solid rich medium. The serial dilutions of overnight cultures of wild-type (WT) or *hcs1*Δ strains (several clones, marked 1–8) in two backgrounds (BY4741 or BY4742) were spotted on the YPD medium supplemented with 4-CP as indicated. The plates were incubated for 5 days at 28 °C. (**b**) The ectopic expression of *HCS1* sensitizes cells to 4-CP. The wild-type (WT) or *hcs1*Δ strains in the BY4741 background carrying the empty ([-]) vector or plasmid with the *HCS1* gene ([*HCS1*]) were incubated overnight in SC-leu medium. The cultures were serially diluted and spotted on the YPD medium with or without 4-CP as indicated. The plates were incubated for 6 days at 28 °C.

**Figure 2 ijms-23-09913-f002:**
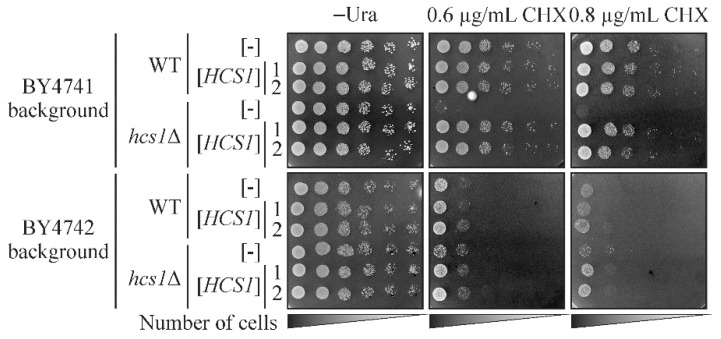
The *hcs1*Δ strain is hypersensitive to cycloheximide. The wild-type (WT) and *hcs1*Δ strains in the BY4741 or BY4742 backgrounds transformed with empty vector ([-]) or plasmid bearing the *HCS1* gene ([*HCS1*]) were grown overnight in SC-ura medium, serially diluted and spotted on SC-ura medium supplemented with cycloheximide (CHX) as indicated. Two independent transformants bearing the *HCS1* gene were tested. The plates were incubated for 6 days at 28 °C.

**Figure 3 ijms-23-09913-f003:**
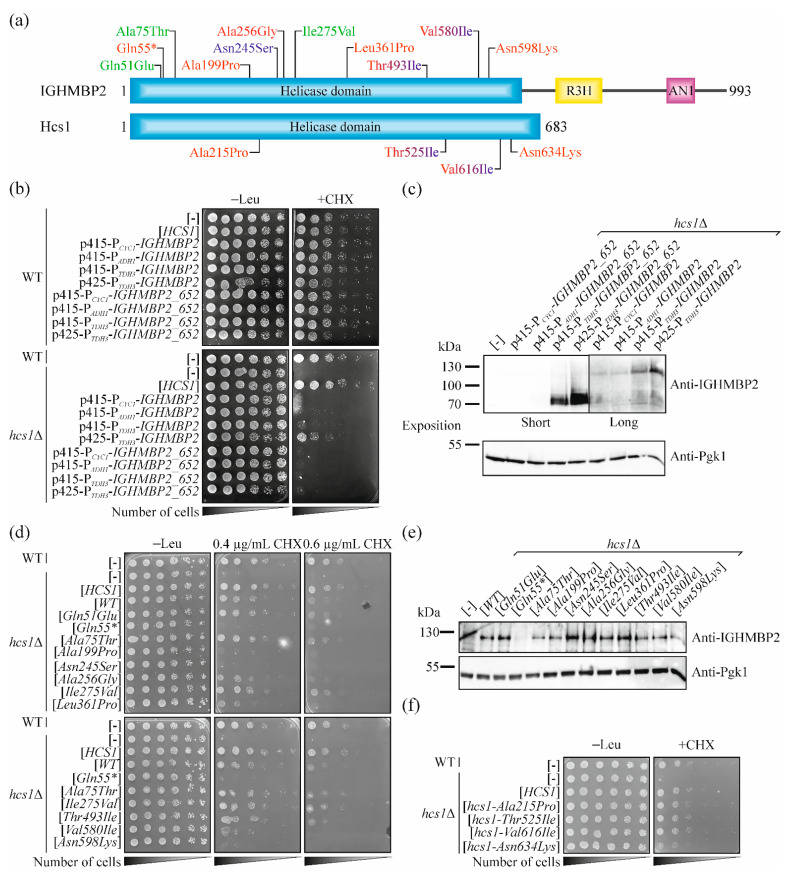
Expression of human *IGHMBP2* cDNA suppresses the growth defect of *hcs1*Δ strains on a cycloheximide-containing medium, while the introduction of pathogenic mutations impair this ability. (**a**) Schematic representation of human IGHMBP2 and yeast Hcs1 domain structure; helicase domain, blue; R3H, yellow; AN1-type zinc finger motif (AN1), pink; amino acid residue substitutions in IGHMBP2 proteins in SMARD1-affected patients, red; amino acid residue substitutions in CMT2S, navy blue; benign changes, green. The p.Thr493Ile and p.Val580Ile found in both SMARD1 and CMT2S patients are indicated in red and navy blue (similar to their equivalents in Hcs1, p.Thr525Ile and p.Val616Ile). Substitutions in IGHMBP2 are indicated above and their equivalents in Hcs1 below scheme. (**b**) Suppression of the growth phenotype of *hcs1*Δ yeast cells by full-length and truncated versions of *IGHMBP2* with different expression levels. Overnight cultures of wild-type (WT) or *hcs1*Δ cells harboring an empty vector ([-]), plasmids with *HCS1* ([*HCS1*]) or different plasmids (p415-P_CYC1_, p415-P_ADH1_, p415-P_TDH3_ and p425-P_TDH3_) with full-length ([*IGHMBP2*]) or truncated versions of *IGHMBP2*, encoding only the helicase domain ([*IGHMBP2_652*]) were diluted to OD600 ≈1. Ten-fold serial dilutions were spotted on SC-leu medium with 0.8 µg/mL (+CHX) or without CHX (-Leu) and incubated for 6 days at 28 °C. (**c**) Level of human full-length and truncated IGHMBP2 proteins in *hcs1*Δ yeast cells. Yeast *hcs1*Δ cells harboring an empty vector ([-]) or plasmids (p415-P_CYC1_, p415-P_ADH1_, p415-P_TDH3_ and p425-P_TDH3_) with the full-length cDNA *IGHMBP2* gene ([*IGHMBP2*]) or only the helicase domain-encoding sequence ([*IGHMBP2_652*]) were cultured, collected and disrupted as described in the Section 4. The total cell extracts obtained were analyzed by SDS-PAGE, followed by immunoblotting with anti-IGHMBP2 or anti-Pgk1 antibodies, as indicated. [WT], wild-type IGHMBP2. (**d**) Suppression of the cycloheximide hypersensitivity of *hcs1*Δ yeast cells by *IGHMBP2* alleles. Ten-fold serial dilutions of overnight cultures of wild-type (WT) or *hcs1*Δ cells harboring an empty vector ([-]) or plasmids with *HCS1* ([*HCS1*]) or *IGHMBP2* alleles, as indicated, were spotted on SC-leu medium with or without CHX and incubated for 5 days. (**e**) Level of IGHMBP2 mutant proteins in *hcs1*Δ yeast cells. Total cell extracts from yeast *hcs1*Δ cells harboring an empty vector ([-]) or plasmids with various I*GHMBP2* variants, as indicated were prepared and analyzed as in (**c**). [WT], wild-type *IGHMBP2*. (**f**) The influence of IGHMBP2 substitutions found in patients on the functioning of Hcs1. Wild-type (WT) or *hcs1*Δ cells harboring an empty vector ([-]) or plasmids with the indicated *HCS1* variants were grown overnight in SC-leu medium, diluted and spotted on SC-leu medium without (-Leu) or with 0.7 µg/mL CHX. Plates were incubated for 5 days at 28 °C.

**Table 1 ijms-23-09913-t001:** Characteristics of tested mutations.

*IGHMBP2* Variant	ClinVar Status	Clinical Phenotype	In Vitro Phenotypes	Functionality in CHX-Hypersensitivity Test	Phenotype Compatibility	Comments
c.151C > G p.Gln51Glu	Benign/Likely benign/Uncertain significance	n.a.	n.t.	Functional protein	Good	
c.163C > T p.Gln55*	Pathogenic	SMARD1	n.t.	Non-functional protein	Good	Compound heterozygosity p.Gln55*/p.Gln657* [47]
c.223G > A p.Ala75Thr	Benign	n.a.	n.t.	Functional protein	Good	
c.595G > A p.Ala199Pro	Uncertain significance	SMARD1	n.t.	Non-functional protein	Good	Compound heterozygosity p.Ala199Pro/p.Ser539_Tyr541del [1]
c.734A > G p.Asn245Ser	Uncertain significance	CMT2S; Independent ambulation	n.t.	Non-functional protein	Medium	Compound heterozygosity, but second allele was not identified [9]
c.767C > G p.Al256Gly	Uncertain significance	SMARD1	n.t.	Partially functional protein	Good	In patient, three different mutations in *IGHMBP2* were identified and confirmed (p.Ala256Gly, p.Ala398Glu and p.Glu514Lys) [49]; also in heterozygous state [58]
c.823A > G p.Ile275Val	Benign	n.a.	n.t.	Functional protein	Good	
c.1082T > C p.Leu361Pro	Pathogenic	Infantile SMARD1 (with pCys496*; p.Leu577Pro; c.1060 + 1G > T; p.GLu382Lys or p.Glu514Lys); juvenile SMARD1 (with p.Thr493Ile)	No ATPase or helicase activity [15]	Partially functional protein	Good	Compound heterozygosity: p.Leu361Pro/p.Cys496* [51] p.Leu361Pro/p.Leu577Pro [50]; p.Leu361Pro/c.1060 + 1G > T [47] p.Leu361Pro/ p.Thr493Ile [5]; p.Leu361Pro/ p.Glu382Lys [52]; p.Leu361Pro/p.Glu514Lys [53]
c.1478C > T p.Thr493Ile	Pathogenic/Likely pathogenic	Infantile SMARD1 (with p.Arg788*; p.Cys496*; p.Leu155Gln; p.Glu514Lys or c.86 + 1022_c.257–191del2894); Juvenile SMARD1 (with p.Leu361Pro); CMT2S (with p.Lys328Thrfs46* but SMARD1 in patient with Kabuki syndrome)	ATPase and helicase activity as in WT; decrease in RNA/DNA binding capacity; tendency to form aggregates and to degrade [5,15]	Functional protein	Medium	Compound heterozygosity: p.Leu361Pro/p.Thr493Ile; pThr493Ile/p.Arg788* [5] p.Thr493Ile/p.Cys496* [54] p.Leu155Gln/p.Thr493Ile [55] p.Thr493Ile/p.Lys328Thrfs46* [56]; p.Thr493Ile/ p.Glu514Lys p.Thr493Ile/c.86 + 1022_c.257–191del2894 [53]
c.1738G > A p.Val580Ile	Likely pathogenic	SMARD1 ((homozygotes and p.Arg785fs; CMT2S (with Pro531Thr)	No ATPase or helicase activity [48]	Partially functional protein	Good	In homozygous state [8,48] and compound heterozygosity: p.Val580Ile/p.Pro531Thr [9]; p.Val580Ile/p.Arg785fs [57]
c.1794C > A p.Asn598Lys	Uncertain significance	SMARD1	n.t.	Non-functional protein	Good	Compound heterozygosity: p.Arg147*/p.Asn598Lys [1]

n.t.—not tested; n.a.—not applicable; WT—wild-type.

**Table 2 ijms-23-09913-t002:** *S. cerevisiae* strains used in this study.

Strain	Genotype	Source
BY4741	*MAT***a** *his3*Δ*1 leu2*Δ*0 met15*Δ*0 ura3*Δ*0*	Open Biosystem
BY4742	MATα *his3*Δ*1 leu2*Δ*0 lys2*Δ*0 ura3*Δ*0*	Open Biosystem
*ccr4*Δ	BY4741 *ccr4*::KanMX	This study
*doa1*Δ	BY4741 *doa1*::KanMX	Open Biosystem
*fap1*Δ	BY4741 *fap1*::KanMX	Open Biosystem
*maf1*Δ	BY4742 *maf1*::KanMX	[61]
*rbs1*Δ	BY4741 *rbs1*::KanMX	[44]
*sqs1*Δ	BY4741 *sqs1*::KanMX	Open Biosystem
*hcs1*Δ1	BY4741 *hcs1*::NatMX	This study
*hcs1*Δ2	BY4742 *hcs1*::NatMX	This study
*ccr4*Δ *hcs1*Δ	BY4741 *ccr4*::KanMX *hcs1*::NatMX	This study
*doa1*Δ *hcs1*Δ	BY4741 *doa1*::KanMX *hcs1*::NatMX	This study
*fap1*Δ *hcs1*Δ	BY4741 *fap1*::KanMX *hcs1*::NatMX	This study
*maf1*Δ *hcs1*Δ	BY4742 *maf1*::KanMX *hcs1*::NatMX	This study
*rbs1*Δ *hcs1*Δ	BY4741 *rbs1*::KanMX *hcs1*::NatMX	This study
*sqs1*Δ *hcs1*Δ	BY4741 *sqs1*::KanMX *hcs1*::NatMX	This study

**Table 3 ijms-23-09913-t003:** Plasmids used in this study.

Plasmid	Source
pRS315 [*CEN6 LEU2*]	[62]
pRS416 [*CEN6 URA3*]	[62]
p415-P*_CYC1_* [*CEN6 LEU2*]	[63]
p415-P*_ADH1_* [*CEN6 LEU2*]	[63]
p415-P*_TDH3_* [*CEN6 LEU2*]	[63]
p425-P*_TDH3_* [2µ *LEU2*]	[63]
pRS315-*HCS1*	This study
pRS416-*HCS1*	This study
pRS315-*HCS1-Ala215Pro*	This study
pRS315-*HCS1-Thr525Ile*	This study
pRS315-*HCS1-Val616Ile*	This study
pRS315-*HCS1-Asn634Lys*	This study
pCMV3-*IGHMBP2*	Sino Biological
p415-P*_CYC1_*-*IGHMBP2_652*	This study
p415-P*_ADH1_*-*IGHMBP2_652*	This study
p415-P*_TDH3_*-*IGHMBP2_652*	This study
p425-P*_TDH3_*-*IGHMBP2_652*	This study
p415-P*_CYC1_*-*IGHMBP2*	This study
p415-P*_ADH1_*-*IGHMBP2*	This study
p415-P*_TDH3_*-*IGHMBP2*	This study
p425-P*_TDH3_*-*IGHMBP2*	This study
p425-P*_TDH3_*-*IGHMBP2-Gln51Glu*	This study
p425-P*_TDH3_*-*IGHMBP2-Gln55**	This study
p425-P*_TDH3_*-*IGHMBP2-Ala75Thr*	This study
p425-P*_TDH3_*-*IGHMBP2-Ala199Pro*	This study
p425-P*_TDH3_*-*IGHMBP2-Asn245Ser*	This study
p425-P*_TDH3_*-*IGHMBP2-Al256Gly*	This study
p425-P*_TDH3_*-*IGHMBP2-Ile275Val*	This study
p425-P*_TDH3_*-*IGHMBP2-Leu361Pro*	This study
p425-P*_TDH3_*-*IGHMBP2-Thr493Ile*	This study
p425-P*_TDH3_*-*IGHMBP2-Val580Ile*	This study
p425-P*_TDH3_*-*IGHMBP2-Asn598Lys*	This study

## Data Availability

The data provided in this study are available on request from J.K.

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
