# Peer review of "Validation of the Pathogenic Effect of *IGHMBP2* Gene Mutations Based on Yeast *S. cerevisiae* Model"

_ijms, 2022, doi:10.3390/ijms23179913_

Round 1
Reviewer 1 Report
Validation of pathogenic effect of IGHMBP2 gene mutations based on yeast S. cerevisiae model
Summary:
IGHMBP2 mutations cause spinal muscular atrophy with respiratory distress type 1 (SMARD1) and CMT2. IGHMBP2, is well defined, but mutations are not always sufficient to be sure of diagnosis as the gene is highly polymorphic and pathogenicity of many gene variants is unknown.
In this study, they analysed the simple yeast model to establish the significance of IGHMBP2 variants for disease. They have shown that cDNA of human gene encodes IGHMBP2 protein which is functional in yeast cells and different pathogenic mutations affect this functionality. The model may be used to quickly distinguish between pathogenic and non-pathogenic mutations identified in IGHMBP2 in correlation with other genetic proof of pathogenicity.
Opinion:
A detailed and careful yeast S. cerevisiae study of IGHMBP2 mutations/variants. The research group are very good and have an excellent tract record in neuromuscular diseases. The IGHMBP2 gene is difficult to work with as polymorphic and large. The yeast work will be an adjust to mutation interpretation.
Points:
Does mutation/variant location on the gene influence function? Or clinical phenotype?
How does mutation pathogenicity in yeast compare with genetic/protein prediction tools such as CADD score etc
The English could be improved a little throughout the paper, English language/editorial team read through.
The table could be clearer on the yeast phenotype vs other features such as clinical phenotype.
The figure could also be clearer with putting the mutation/variant on the figure with the effect.
Author Response
The answer is attached file

Reviewer 2 Report
In this paper Rzepnikowska and coworked set up a simple and fast approach to screen for genetic variants of the IGHMBP2 gene, involved in the SMARD1 disease, by using S. cerevisiae.
Given the high degree of polymorphisms of this gene, a fast screening of genetic variants would be very useful and could have potential clinical outcomes.
The idea of using a complementation assay in yeast to evaluate the effect of mutations in the human protein is interesting. The problem is that the screening is all based on cycloheximide sensitivity, which however cannot be related to the function of Hcs1 protein. Why is hcs1Delta strain more sensitive to CHX? Without a reasonable explanation that connects CHX-sensitivity to protein function, I cannot really believe in the strength of this approach.
Minor issues:
-Please remove lines 100-102
-Supplementary Fig. S1 should not be in the main article but in a different file
-Figures with spots have molds, please replace those figures with other (I am sure that the authors repeated the experiments several times, so they can replace the images)
-The authors should explain the discrepancy between what they have found and reference 36, as concerns sensitivity to 4CP of hcs1Delta strain
-line 103 remove the word “rather”
-English should be revised
Round 2
Reviewer 2 Report
The authors have fixed some the minor issues raised during the revision, but the main flaw of the paper is still present. The authors claim to have preliminary results about a possible involvement of Hcs1 in translation. Their preliminary data about polysomes should be repeated and included in the manuscript, to support the use of CHX for their screening. Otherwise, the paper cannot be accepted.
Author Response
The answer is in attached file.

Round 3
Reviewer 2 Report
I really understand the problems of the authors concerning fundings and personnel, these are very common problems unfortunately. I am still convinced about the utility of adding data to sustain the use of CHX for the screening, as I have told during the first and second revision. The paper would be much more solid, and the proposed screening could have a stronger translational potential. However, considering that this would be the third round of revision, I am also convinced that the Editor does not want to reject your paper. So, I leave the decision to the Editor.
Author Response
Thank you for your decission.